# Antimicrobial susceptibility and risk factors for resistance among *Escherichia coli* isolated from canine specimens submitted to a diagnostic laboratory in Indiana, 2010–2019

**John E. Ekakoro**[1], **G. Kenitra Hendrix**[2], **Lynn F. Guptill**[3], **Audrey Ruple**[4]*

**1** Department of Public and Ecosystem Health, College of Veterinary Medicine, Cornell University, Ithaca, NY, United States of America, **2** Department of Comparative Pathobiology, College of Veterinary Medicine, Purdue University, West Lafayette, IN, United States of America, **3** Department of Veterinary Clinical Sciences, College of Veterinary Medicine, Purdue University, West Lafayette, IN, United States of America, **4** Department of Population Health Sciences, Virginia-Maryland College of Veterinary Medicine, Virginia Tech, Blacksburg, VA, United States of America

* aruple@vt.edu

**Data Availability Statement:** All data are available here: https://doi.org/10.7294/19358228.

**Funding:** The Integrative Data Science Initiative at Purdue University (https://www.purdue.edu/data-

## Abstract

*Escherichia coli* (*E. coli*) is the most common Gram-negative pathogen isolated in human infections. Antimicrobial resistant (AMR) *E. coli* originating from dogs may directly or indirectly cause disease in humans. The objective of this study was to calculate the proportion of antimicrobial susceptible *E. coli* isolated from canine specimens submitted to the Indiana Animal Disease Diagnostic Laboratory and to identify temporal patterns of susceptibility among these isolates. Susceptibility data of 2,738 *E. coli* isolates from dogs from 2010 through 2019 were used in this study. Proportions of isolates susceptible to the various antimicrobials were calculated using SAS statistical software and the Cochran-Armitage trend test was used to investigate the temporal trends in susceptibility. A multivariable binary logistic regression model was built to investigate the association between host factors and AMR. Overall, 553/2,738 (20.2%) of the isolates were susceptible to 17 of the 27 antimicrobials examined. Of the 2,638 isolates examined for amikacin susceptibility, 2,706 (97.5%) were susceptible, 2,657/2,673 (99.4%) isolates were susceptible to imipenem, and 2,099/2,670 (78.6%) were susceptible to marbofloxacin. A significant decreasing trend in susceptibility was observed for amoxicillin-clavulanic acid ($P$<0.0001), ampicillin ($P$<0.0001), Cefazolin ($P$<0.0001), ceftazidime ($P$ = 0.0067), chloramphenicol ($P$<0.0001), and orbifloxacin ($P$ = 0.008). The overall percentage of AMR isolates (isolates not susceptible to at least one antimicrobial) was 61.7% (1,690/2,738) and 29.3% (801/2,738) of isolates were multidrug resistant. Multivariable regression analyses showed significant associations between AMR and age ($P$ = 0.0091), breed ($P$ = 0.0008), and sample isolation site/source ($P$<0.0001). The decreasing trend in the proportion of isolates susceptible to several beta-lactam antimicrobials suggests that resistance of *Escherichia coli* in dogs to these antimicrobials could be increasing in Indiana. The decreasing trend in susceptibility to these drugs could be due to selection pressure from antimicrobial use.

science/) provided funding to AR for this work and JE was provided salary using these funds. The funders had no role in study design, data collection and analysis, decision to publish, or preparation of the manuscript.

**Competing interests:** The authors have declared that no competing interests exist.

## Introduction

*Escherichia coli*, a member of the ESBL-producing Enterobacteriaceae, is the most common Gram-negative pathogen isolated in human clinical infections, and antimicrobial resistant (AMR) *E. coli* pose a threat to both human and animal health [1]. Previous studies have reported isolation of transmissible AMR *E. coli* in dogs [2]. *E. coli* is the most common cause of urinary tract infections in humans and dogs and sharing of *E. coli* strains between dogs and humans can occur [3]. The CDC reported that an estimated 197,400 cases of and 9,100 deaths occurred due to ESBL-Enterobacteriaceae infections among hospitalized patients in 2017 in the US [4]. AMR *E. coli* originating from dogs may directly or indirectly cause disease in humans [5].

However, we do not know the total number of cases in which AMR *E. coli* cause disease or death in dogs in the US. Without this knowledge, we cannot fully understand the role dogs may play in spreading AMR *E. coli* infections to humans. In addition, understanding the patterns of antimicrobial susceptibility of bacterial isolates identified from dogs is a critical step in antimicrobial stewardship and in the containment of AMR within the One Health framework. The objectives of this study were to: 1) calculate the proportion of antimicrobial susceptible E. coli isolates identified in canine specimens submitted to the Indiana Animal Disease Diagnostic Laboratory (ADDL) from January 1, 2010, through December 1, 2019; 2) identify temporal trends in susceptibility among these isolates to individual antimicrobials tested; and 3) to identify the temporal patterns and host risk factors for AMR and multidrug resistance (MDR) among these isolates.

## Materials and methods

### Source of data and ethical approval

The study was exempted from oversight by the Purdue University Institutional Animal Care and Use Committee (IACUC). We used secondary data obtained from the Indiana ADDL and informed consent was not required. No field studies or experiments were conducted in this study, and the study did not directly involve use of animals and posed no risk to clients (animal owners). Data from *E. coli* isolates phenotypically assessed for AMR from January 1, 2010, through December 31, 2019, were utilized. The variables extracted from the dataset included: the age of the dog, breed, sex, geographic location (localized to zip code) of its home, and host source (anatomic location) of isolation of the pathogen.

The antimicrobial susceptibility test (AST) results used in this analysis were obtained using the broth microdilution method using the Sensititre™ Companion Animal Gram Negative COMPGN1F Vet AST Plates purchased from ThermoFisher scientific-USA, the Mueller-Hinton broth as the media, and *Escherichia coli* (ATCC® 25922™) as the quality control strain. All testing was in accordance with the ADDL standard operating procedure for broth microdilution method. This yielded quantitative data (minimum inhibitory concentration) and the isolates were categorized as susceptible (S), intermediate (I), or resistant (R) based upon Clinical and Laboratory Standards Institute (CLSI) guidelines that were current at the time the isolate was tested [6]. The susceptibility testing was performed for 35 drugs: amikacin, amoxicillin, ampicillin, azithromycin, cefazolin, cefovecin, cefoxitin, cefpodoxime, ceftazidime, ceftiofur, chloramphenicol, chlortetracycline, clarithromycin, clindamycin, danofloxacin, doxycycline, enrofloxacin, erythromycin, florfenicol, gentamicin, imipenem, marbofloxacin, neomycin, oxacillin, oxytetracycline, penicillin, rifampin, spectinomycin, sulfadimethoxine, tetracycline, tiamulin, ticarcillin, ticarcillin-clavulanate, tilmicosin, trimethoprim, tulathromycin, and tylosin. Drugs with complete susceptibility data or with more than 500 isolates tested were considered in these analyses.

Overall, 27 antimicrobials from 10 antimicrobial classes were included in the final analyses. The antimicrobial classification conformed with the classification described by Riviere and Papich [7] and the 10 classes included aminoglycosides, the penicillins, cephalosporins and cephamycins, carbapenems, amphenicols, fluoroquinolones, macrolides, lincosamides, tetracyclines, and antifolate. All 10 classes belonged to either critically important antimicrobial classes for human medicine (e.g. aminoglycosides, carbapenems, penicillins) or highly important antimicrobials (e.g. amphenicols, antifolate) as classified by the World Health Organization (WHO) [8]. For AMR and MDR determination, drugs known to exhibit intrinsic resistance phenotypes in Enterobacteriaceae [9] (e.g. penicillin, oxacillin, clindamycin, and erythromycin) were excluded.

## Data and statistical analysis

Data cleaning and preparation was performed in Microsoft Excel. The data were assessed for completeness, duplicates were removed, and only complete records were included in the analyses. Geographic origins of the samples located to zip code were categorized at the county and state spatial scales. The state spatial scale categories were further grouped into within Indiana, out-of-state, and unknown (for those where no geographic origin was reported). The sex of the dog was categorized as male, female, or intersex regardless of neuter status. Age was categorized into seven age groups: less than 1 year, 1 to 3 years, over 3 to 6 years, over 6 to 8 years, over 8 to 10years, over 10 to 12years, and greater than 12 years of age as described previously [10]. We removed one case from the age category due to an implausible age designation of 95 years.

Dog breeds were grouped based on the American Kennel Club (AKC) breed group classification as described by Conner and colleagues [11]. However, three breeds (English shepherd, Jack Russel terrier, and Pitbull) that were not listed on the AKC grouping system were classified based on the United Kennel Club (UKC) grouping [12]. Dogs identified in the dataset as mixed breed were treated as such in the final grouping. Two breeds (goldendoodle and cockapoo) that were not yet recognized by any major kennel club were included in the category mixed. If an animal was identified using a non-specific breed name such as poodle, or schnauzer, they were categorized as unknown breed. If breed, sex, or age of the dog was not reported and other data was otherwise complete, it was categorized as "unknown" for the specific category.

The anatomic location or specimen source was categorized as: abdominal cavity/fluid, ear and ocular, feces, respiratory tract, skin, urine and bladder, uterus, vagina and vulva, wounds, and "all others." The "all others" contained specimen sources with very small counts or those with non-specific identities such as fluid, swabs, tissue etc. All AST results reported as "NI" (no interpretation) were excluded from the analysis. A more conservative approach for categorization of all AST data reported as susceptible, intermediate, or resistant was adopted for this study as previously suggested by Sweeney and others [13] and Magiorakos and others [14]. Briefly, the AST data were grouped into two categories "susceptible" and "not susceptible." The "not susceptible" category included the resistant and/or intermediately susceptible isolates. Isolates that were not susceptible to at least one antimicrobial drug were considered to be AMR isolates [11] and isolates that were not susceptible to at least one antimicrobial drug in at least three antimicrobial classes were considered to be MDR as previously described [13]. The CLSI guidelines were used in the analysis of the AST results [15].

**Descriptive analyses.** Statistical analyses were performed in a SAS commercial statistical software. Frequencies and proportions were used to summarize the data. The Cochran-Armitage trend test was used to investigate the temporal trends in the data.

**Univariable and multivariable analysis.** Isolates from intersex dogs and from dogs belonging to the foundation stock service breed group were excluded from the univariable and

multivariable analyses due to small counts. Univariable binary logistic regression was used to investigate the association between geographic origin of sample and AMR. A further analysis of the associations between host factors (age, sex, and breed of the dog, specimen source/type and AMR/MDR) were conducted only for samples with a known in-state address. Variables with a *p*-value ≤ 0.15 in the univariable analysis were considered for inclusion in the multivariable model building. A multivariable binary logistic regression model was built to investigate the association between host factors and AMR. The backward elimination procedure was used to build the multivariable model and only statistically significant predictors (*P*≤ 0.05) were retained in the final main effects multivariable model. In the final model, two-way interactions between age and breed were assessed based on biological plausibility and standard multiple pairwise comparisons were obtained using the SAS "LSMEANS" statement. The model fit was assessed using The Hosmer and Lemeshow Goodness-of-Fit Test. Cluster analysis to discern the spatial patterns of AMR/MDR was deemed untenable due to small sample sizes in the different counties in Indiana.

## Results

### Sample characteristics

A total of 2,738 *E. coli* isolates were included in the general analysis of these data. Of these, 1,641 (59.9%) were isolated from samples obtained from female dogs, 881 (32.2%) from male dogs, three (0.1%) were from intersex dogs, and 190 (7%) samples were from dogs that did not have sex identified. Most of the samples (n = 2,058; 75.2%) were identified using an in-state zip code while 275 (10%) were identified as being from out-of-state samples; 405 (14.8%) samples had no geographic origin reported. Out-of-state samples came from 18 states: Illinois (n = 175), Michigan (n = 23), Ohio (n = 23), Maryland (n = 10), Tennessee (n = 9), Missouri (n = 5), Georgia (n = 5), West Virginia (n = 5), California (n = 4), Kentucky (n = 4), Florida (n = 3), Texas (n = 2), Pennsylvania (n = 2), Virginia (n = 1), Wisconsin (n = 1), Nebraska (n = 1), Alabama (n = 1), and Arkansas (n = 1) (Table 1).

### Proportions and trends in susceptibility to different antimicrobials

Overall, 553 (20.2%) of the isolates were susceptible to 17 of the 27 antimicrobials examined. *E. coli* susceptibility to marbofloxacin was 78.6% (2,099/2,670) and ranged from 83.3% (170/204) susceptible isolates tested in 2010 to 75.7% (234/309) susceptible isolates tested in 2019. Overall susceptibility to doxycycline was 74.4% (1,999/2,688) and ranged from 77.5% (158/204) susceptible isolates tested in 2010 to 72.5 (227/313) susceptible isolates tested in 2019 (Table 2). Statistically significant temporal trends were observed among 10 of the 27 antimicrobials evaluated (Table 2). A significant (*P* < 0.05) downward (decreasing) trend in susceptibility was observed for amoxicillin-clavulanic acid, ampicillin, cefalexin, cefazolin, ceftazidime, cephalothin, chloramphenicol, and orbifloxacin (Table 2).

### Antimicrobial resistance (AMR) and multi-drug resistance (MDR)

The overall percentage of AMR (isolates not susceptible to at least one antimicrobial) in isolates was 61.7% (n = 1,690) and 29.3% (801) of isolates were MDR. Of the 1,690 AMR isolates, 47.4% (801/1,690) were MDR (Table 3). A significant (*P* = <0.0001) upward trend in AMR was observed while MDR significantly (*P* = 0.0083) decreased (Fig 1). Geographic region of sample origin (e.g., out-of-state versus in-state) was significantly associated with AMR (*P* < .0001). The odds of an isolate being shown to have resistance to at least one antimicrobial were two times higher in all (combined) out-of-state samples when compared to samples from

**Table 1. Characteristics of all *Escherichia coli* isolates tested for antimicrobial susceptibility at the Indiana Animal Disease Diagnostic Laboratory, from January 2010 to December 2019.**

| Sample characteristics | Number (%) of isolates |
| --- | --- |
| **Geographic origin of sample** | **N = 2,738** |
| Indiana | 2,058 (75.2) |
| Out-of-state | 275 (10) |
| Location not recorded | 405 (14.8) |
| **Sex** | **N = 2,738** |
| Female | 1,641 (59.9) |
| Male | 881 (32.2) |
| Intersex | 3 (0.1) |
| Unknown | 213 (7.8) |
| **Age of dog (years)** | **N = 2,737** |
| <1year | 208 (7.6) |
| 1-3years | 265 (9.7) |
| >3-6years | 440 (16.1) |
| >6-8years | 413 (15.1) |
| >8-10years | 496 (18.1) |
| >10-12years | 447 (16.3) |
| >12years | 408 (14.9) |
| Unknown | 60 (2.2) |
| **Breed Group** | **N = 2,738** |
| Mixed breed | 583 (21.3) |
| Sporting | 565 (20.6) |
| Working | 312 (11.4) |
| Hound | 256 (9.4) |
| Terrier | 256 (9.4) |
| Toy | 252 (9.2) |
| Herding | 222 (8.1) |
| Non-Sporting | 200 (7.3) |
| Unknown | 88 (3.2) |
| Foundation Stock Service | 4 (0.2) |
| **Isolation source** | **N = 2,738** |
| Abdominal cavity and fluid | 77 (2.8) |
| Ear and Ocular | 138 (5) |
| Feces | 170 (6.2) |
| Respiratory tract | 101 (3.7) |
| Skin | 45 (1.6) |
| Urine and bladder | 1676 (61.2) |
| Uterus, vagina, and vulva | 59 (2.2) |
| Wounds | 71 (2.6) |
| All others | 401 (14.7) |
| **Year of sample collection** | **N = 2,738** |
| 2010 | 206 (7.5) |
| 2011 | 249 (9.1) |
| 2012 | 228 (8.3) |
| 2013 | 232 (8.5) |
| 2014 | 280 (10.2) |
| 2015 | 257 (9.4) |

*(Continued)*

**Table 1.** (Continued)

| Sample characteristics | Number (%) of isolates |
|---|---|
| 2016 | 310 (11.3) |
| 2017 | 294 (10.7) |
| 2018 | 355 (13) |
| 2019 | 327 (12) |

Indiana (OR: 2.04, 95% CI: 1.54–2.7) and the odds of an isolate being shown to have resistance to at least one antimicrobial were 1.89 times higher among samples of unreported (unknown) origin when compared to known Indiana samples (OR: 1.89, 95% CI:1.5–2.39).

**Host factors associated with AMR/MDR in Indiana.** For all samples from known Indiana addresses, 1,191/2,050 (58.1%) were resistant to at least one antimicrobial and 859/2,050 (41.9%) were not resistant to any antimicrobials. Of the 1,191 AMR isolates, 532 (44.7%) were MDR (Table 4).

**Univariable logistic regression.** There was no significant unadjusted association between sex and the outcome of AMR, however breed, age, and isolation source had significant associations with AMR (Table 5). There were no significant unadjusted associations between the four host factors and MDR (Table 6).

**Adjusted associations.** All host factors found to be widely significantly associated ($P \leq$ 0.15) with AMR in the univariable logistic regression models were included in the multivariable logistic regression analyses. Thus, for AMR, age ($P = 0.0149$), breed ($P = 0.0007$) and sample source/sample type ($P < .0001$) were included in the multivariable model. All three host factors were retained in the final multivariable model (Table 7) which showed significant associations between AMR and age ($P = 0.009$), breed ($P = 0.0007$), and sample isolation site/source ($P < 0.0001$). The Hosmer and Lemeshow Goodness-of-Fit Test showed that this model best fit these data ($\chi^2 = 8.05$, DF = 8, $P = 0.429$). The multivariable model showed that controlling for breed and specimen source, the odds of AMR in isolates from dogs aged 1 to 3 years were 1.63 times as high as the AMR odds in isolates from dogs aged between 6 and 8 years and isolates from dogs aged greater than 10 years were more likely to be antimicrobial resistant than those isolated from other age groups. Based on the non-significant unadjusted associations (using a liberal $\alpha = 0.15$), a multivariable model for the association between the host factors and MDR was not built.

## Discussion

In the present study, we found significant trends in susceptibility, total AMR and MDR in canine *E. coli* isolates, and we identified significant associations between AMR and dog age, breed, and the source of the specimens. We found significant declines in the susceptibility to cefalexin, cefazolin, and cephalothin which are 1st generation cephalosporins and to cefpodoxime and ceftazidime which are 3rd generation cephalosporins. Similar to our study, a previous study found high level resistance to commonly used beta lactams (penicillins, cephalosporins) in dogs in the United States [16]. Particularly, 39.7% of all the isolates in the present study were not susceptible to amoxicillin-clavulanic acid and 52.3% were not susceptible to ampicillin, and susceptibility to these drugs significantly declined over time. Similar to our findings, a previous study by Thungrat and others reported high-level resistance (45%) to amoxicillin-clavulanic acid and 52.7% to ampicillin among *E. coli* isolated from dogs in the United States [16]. It is important to note that amoxicillin-clavulanic acid is the most commonly prescribed antimicrobial in many veterinary practices [17–19] and ampicillin is also commonly used to

**Table 2. Trends in antimicrobial susceptibility of *Escherichia coli* isolated from dog specimens tested at the Indiana Animal Disease Diagnostic Laboratory, 2010–2019.**

| Antimicrobial class | Antimicrobial | Percentage (number of specimens tested) of susceptible isolates to an antimicrobial | | | | | | | | | | Total | Statistic (Z)- CAT-T | P-values (CAT-T) |
|---|---|---|---|---|---|---|---|---|---|---|---|---|---|---|
| | | 2010 | 2011 | 2012 | 2013 | 2014 | 2015 | 2016 | 2017 | 2018 | 2019 | | | |
| Aminoglycosides | | | | | | | | | | | | | | |
| | Amikacin | 97.6 (204) | 98.8 (248) | 95.6 (226) | 96.1 (232) | 93.9 (277) | 98.1 (257) | 97.1 (310) | 100 (289) | 99.2 (354) | 97.3 (309) | 97.5 (2706) | -2.1528 | 0.0157 |
| | Gentamycin | 86.4 (206) | 93.6 (249) | 84.7 (228) | 83.2 (232) | 87.9 (280) | 90.3 (257) | 89.4 (310) | 92.9 (294) | 84.2 (355) | 89.3 (327) | 88.2 (2738) | -0.3426 | 0.3660 |
| Amphenicols | Chloramphenicol | 89.2 (203) | 91.1 (248) | 83.2 (226) | 80.6 (232) | 86.3 (277) | 80.2 (257) | 83.9 (310) | 82.7 (289) | 75.5 (351) | 78.8 (217) | 82.8 (2610) | 4.8084 | < .0001 |
| Antifolate | Trimethoprim | 82 (206) | 86.8 (249) | 75.4 (228) | 75.9 (232) | 76.8 (280) | 81.3 (257) | 81.9 (310) | 83.6 (293) | 74.7 (348) | 78.5 (311) | 79.6 (2714) | 1.2911 | 0.0983 |
| Carbapenem | Imipenem | 99 (204) | 100 (248) | 99.1 (226) | 99.6 (230) | 98.9 (275) | 100 (256) | 99.7 (306) | 99.3 (283) | 99.7 (336) | 98.7 (309) | 99.4 (2673) | 0.4271 | 0.3346 |
| Cefalosporin/ Cefamycin | | | | | | | | | | | | | | |
| | Cefalexin | - | - | - | - | - | - | 63.5 (63) | 78.3 (281) | 61.5 (327) | 66 (300) | 67.9 (971) | 2.1955 | 0.0141 |
| | Cefazolin | 74.3 (202) | 75.8 (248) | 73 (226) | 68.5 (232) | 75.1 (277) | 73.5 (257) | 59.1 (308) | 69 (284) | 54.6 (339) | 51.4 (313) | 66.4 (2686) | 8.1388 | < .0001 |
| | Cefovecin | 75 (204) | 77 (248) | 72.1 (226) | 69.6 (230) | 78.2 (275) | 75.8 (256) | 72.9 (306) | 84.4 (282) | 67.4 (331) | 68.9 (309) | 74 (2667) | 1.4236 | 0.0773 |
| | Cefoxitin | 76.5 (204) | 79.8 (248) | 74.3 (226) | 72.6 (230) | 80.7 (275) | 82.8 (256) | 77 (243) | 0 | 0 (1) | 0 | 77.84 (1683) | -0.8763 | 0.1904 |
| | Cefpodoxime | 74 (204) | 76.2 (248) | 71.7 (226) | 71.3 (230) | 77.8 (275) | 75.4 (256) | 71.9 (306) | 84.1 (283) | 66.7 (336) | 67.6 (309) | 73.5 (2673) | 1.6614 | 0.0483 |
| | Ceftazidime | - | - | - | - | - | - | 85.7 (63) | 89.7 (281) | 82.3 (327) | 81.3 (300) | 84.4 (971) | 2.4729 | 0.0067 |
| | Ceftiofur | 75.2 (206) | 74.3 (249) | 71.5 (228) | 66.4 (232) | 75.7 (280) | 73.5 (257) | 73.3 (247) | 72.7 (11) | 85.7 (21) | 79 (19) | 73.1 (1750) | -0.3796 | 0.3521 |
| | Cephalothin | - | 76.5 (115) | 60.2 (226) | 51.1 (141) | - | - | - | 0 (2) | 0 (9) | 7.7 (13) | 58.7 (506) | 6.7500 | < .0001 |
| Penicillins | | | | | | | | | | | | | | |
| | Amoxiclav | 72.6 (204) | 67.2 (137) | 100 (2) | 71.4 (91) | 69.5 (275) | 76.2 (256) | 65.6 (299) | 48.1 (283) | 46.4 (336) | 44.4 (288) | 60.3 (2171) | 9.3130 | < .0001 |
| | Ampicillin | 59.2 (206) | 55.4 (139) | 50 (2) | 55.3 (94) | 53.6 (278) | 57.8 (256) | 50.7 (306) | 37.2 (288) | 37.1 (337) | 38.2 (275) | 47.7 (2183) | 7.1012 | < .0001 |
| | Penicillin | 0 (206) | 0 (247) | 0 (228) | 0 (229) | 0 (276) | 0 (256) | 0 (243) | 0 (7) | 0 (12) | 0 913) | 0 (1717) | - | - |
| | Oxacillin | 0.5 (204) | 0.8 (248) | 2.2 (226) | 1.3 (230) | 1.5 (275) | 0 (256) | 1.7 (243) | 0 (2) | 0 (10) | 7.7 (13) | 1.2 (1707) | -0.6857 | 0.2465 |
| | Piperacillin tazobactam | - | - | - | - | - | - | 100 (63) | 96.4 (281) | 97 (326) | 97.3 (300) | 97.1 (970) | 0.2269 | 0.4103 |
| | Ticarcillin | 60.8 (204) | 58.1 (248) | 54.4 (226) | 52.2 (232) | 54.9 (277) | 58.4 (257) | 63.2 (247) | 83.3 (6) | 52.6 (19) | 72.2 (18) | 57.6 (1734) | -0.9315 | 0.1758 |
| | Ticarcillin Clav | 72.6 (204) | 70.2 (248) | 70.8 (226) | 64.4 (230) | 65.5 (275) | 70.3 (256) | 67.9 (243) | 0 | 0 (1) | 0 | 68.6 (1683) | 1.2077 | 0.1136 |
| Fluoroquinolones | | | | | | | | | | | | | | |
| | Enrofloxacin | 83 (206) | 80.3 (249) | 74.1 (228) | 73 (230) | 79.5 (278) | 78.9 (256) | 76.8 (306) | 91.7 (266) | 73.3 (326) | 73.1 (309) | 78.2 (2654) | 1.0780 | 0.1405 |
| | Marbofloxacin | 83.3 (204) | 81.9 (248) | 74.3 (226) | 74.4 (230) | 80.4 (275) | 78.9 (256) | 77.5 (306) | 88.3 (282) | 73.1 (334) | 75.7 (309) | 78.6 (2670) | 1.2731 | 0.1015 |

*(Continued)*

**Table 2.** (Continued)

| Antimicrobial class | Antimicrobial | Percentage (number of specimens tested) of susceptible isolates to an antimicrobial | | | | | | | | | | Total | Statistic (Z)- CAT-T | P-values (CAT-T) |
|---|---|---|---|---|---|---|---|---|---|---|---|---|---|---|
| | | 2010 | 2011 | 2012 | 2013 | 2014 | 2015 | 2016 | 2017 | 2018 | 2019 | | | |
| | Orbifloxacin | - | - | - | - | - | - | 71.4 (63) | 85.4 (280) | 72.3 (325) | 73 (300) | 76.2 (968) | 2.3941 | 0.0083 |
| Lincosamide | Clindamycin | 0 (206) | 0 (249) | 0 (228) | 0 (230) | 0 (278) | 0 (256) | 0 (243) | 0 (7) | 0 (12) | 0 (14) | 0.06 (1723) | -2.7964 | 0.0026 |
| Macrolide | Erythromycin | 0 (204) | 0 (248) | 0 (226) | 0 (232) | 0 (277) | 0 (257) | 0 (202) | - | - | - | 0 (1646) | - | - |
| Tetracyclines | | | | | | | | | | | | | | |
| | Doxycycline | 77.5 (204) | 76.6 (248) | 72.1 (226) | 68.4 (231) | 75.8 (277) | 73.4 (256) | 79.2 (307) | 75.9 (286) | 72.1 (340) | 72.5 (313) | 74.4 (2688) | 0.5936 | 0.2764 |
| | Tetracycline | - | - | - | - | - | - | 81 (63) | 74.4 (285) | 70 (327) | 72.1 (301) | 72.6 (976) | 1.3344 | 0.0910 |

treat bacterial infections in dogs [16]. Therefore, the decreasing trend in the proportion of isolates susceptible to antimicrobials in the beta lactam group in this study could be due to selection pressure from antimicrobial use. For the fluoroquinolone drugs, 21.8% of all the isolates tested were not susceptible to enrofloxacin. A previous study conducted in the northeastern US reported that nearly 20% of the *E. coli* isolated from dogs during the period 2004–2011 were resistant to enrofloxacin [20]. Also, among the fluoroquinolone antimicrobials, the decline in susceptibility to orbifloxacin observed could be associated with selection pressure from antimicrobial use.

The level of AMR in *E. coli* is a good indicator of AMR in bacterial pathogens of dogs and other species [21, 22] because of its ubiquitous nature and its ability to act as a reservoir of AMR genes that can transferred to other pathogens through horizontal gene transfer [23]. Additionally, AMR in *E. coli* is suggested to be a good sentinel of the effects of selective pressure from AMU [24]. Therefore, the significant increase in AMR *E. Coli* observed in this study could be an indicator of an increasing AMR trend among other pathogenic bacteria in the dog populations served by this diagnostic laboratory. This suggests for a need for more concerted efforts in controlling AMR in small animal practice through judicious AMU. The decreasing trend observed for MDR could have resulted from the varying susceptibility trends observed for individual antimicrobials where some individual drugs had decreasing susceptibility trends while others had increasing susceptibility. Corner and others attributed similar decreases in MDR in *Staphylococcus spp.* to variability in individual drug susceptibility [11].

The total lack of susceptibility to clindamycin and erythromycin observed is due to intrinsic resistance [9]. Enterobacteriaceae such as *E. coli* are known to be intrinsically resistant to lincosamides and macrolides such as clindamycin and erythromycin respectively. This information

**Table 3. Trends in antimicrobial resistance and multidrug resistance among *Escherichia coli* isolated from dog specimens at the Indiana Animal Disease Diagnostic Laboratory, 2010–2019.**

| | Percentage (number of specimen tested) of AMR/MDR isolates | | | | | | | | | | Total | Statistic (Z)- CAT-T | P-values (CAT-T) |
|---|---|---|---|---|---|---|---|---|---|---|---|---|---|
| | 2010 | 2011 | 2012 | 2013 | 2014 | 2015 | 2016 | 2017 | 2018 | 2019 | | | |
| AMR | 49 (206) | 50.2 (249) | 59.2 (228) | 63.8 (232) | 57.1 (280) | 54.5 (257) | 60 (310) | 72.1 (294) | 70.7 (355) | 71 (327) | 61.7 (2738) | -7.4123 | < .0001 |
| MDR | 48.5 (101) | 52.8 (125) | 56.3 (135) | 53.4 (148) | 43.8 (160) | 49.3 (140) | 45.2 (186) | 39.2 (212) | 48.6 (251) | 44.4 (232) | 47.4 (1690) | 2.3959 | 0.0083 |

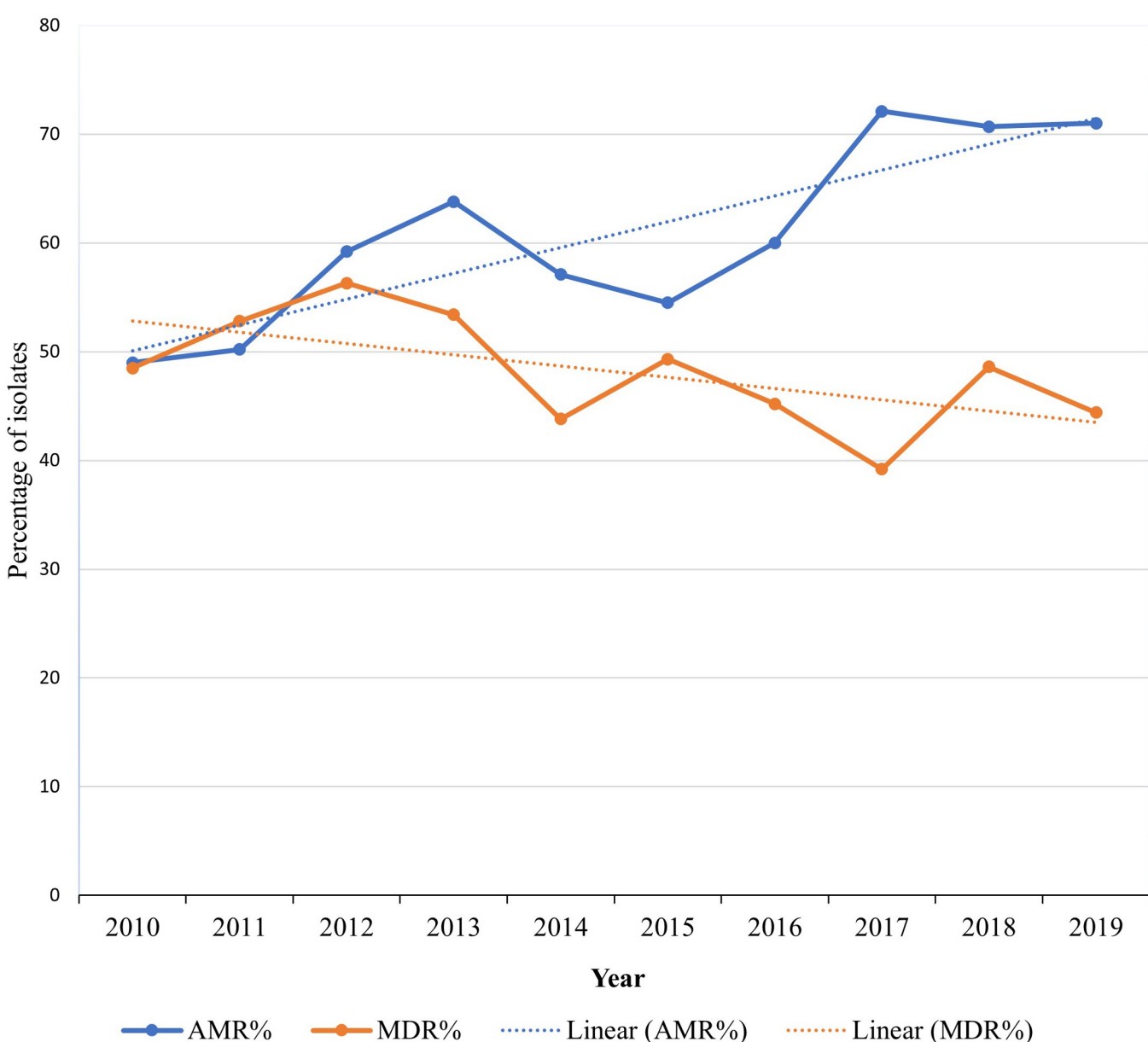

**Fig 1. A graphical representation of the temporal trends in antimicrobial resistance and multidrug resistance among *Escherichia coli* isolated from dog specimens at the Indiana Animal Disease Diagnostic Laboratory, 2010–2019.**

is provided here to guide veterinary clinicians who might find it useful when deciding which antimicrobial to select.

We found high susceptibility of the isolates to amikacin (97.6% susceptibility in 2010 and 97.3% in 2019) and observed a significant increase in susceptibility to this drug. Similar to our findings, a previous study that investigated the antimicrobial susceptibility patterns of *E. coli* in dogs and cats in the United States found only 0.7% of 2,390 canine *E. coli* isolates were resistant to amikacin [16]. Another study in Canada found 93.8% of 3,364 canine *E. coli* isolates were susceptible to Amikacin [18]. The high susceptibility and increasing trend in susceptibility to amikacin observed in the present study could be indicative of limited use of this antimicrobial in small animal practice in Indiana. The limited use of this drug could be associated with

**Table 4. The distribution of isolates from Indiana based on host factors and AMR status.**

| Host factors | Total number (%) of isolates assessed for AMR | Number (%) of AMR isolates | | Total number (%) of isolates assessed for MDR | Number (%) of MDR isolates | |
|---|---|---|---|---|---|---|
| | | No | Yes | | No | Yes |
| **Sex** | **N = 2050** | | | **N = 1191** | | |
| Female | 1239 (60.4) | 509 (24.8) | 730 (35.6) | 730 (61.3) | 394 (33.1) | 336 (28.2) |
| Male | 617 (30.1) | 265 (12.9) | 352 (17.2) | 352 (29.6) | 200 (16.8) | 152 (12.8) |
| Unknown | 194 (9.5) | 85 (4.2) | 109 (5.3) | 109 (9.2) | 65 (5.5) | 44 (3.7) |
| **Age (years)** | **N = 2050** | | | **N = 1191** | | |
| <1year | 177 (8.6) | 78 (3.8) | 99 (4.8) | 99 (8.3) | 50 (4.2) | 49 (4.1) |
| 1-3years | 209 (10.2) | 71 (3.5) | 138 (6.7) | 138 (11.6) | 71 (6) | 67 (5.6) |
| >3-6years | 319 (15.6) | 137 (6.7) | 182 (8.9) | 182 (15.3) | 105 (8.8) | 77 (6.5) |
| >6-8years | 330 (16.1) | 161 (7.9) | 169 (8.2) | 169 (14.2) | 95 (8) | 74 (6.2) |
| >8-10years | 376 (18.3) | 166 (8.1) | 210 (10.2) | 210 (17.6) | 132 (11) | 78 (6.6) |
| >10-12years | 310 (15.1) | 112 (5.5) | 198 (9.7) | 198 (16.6) | 108 (9) | 90 (7.6) |
| >12years | 279 (13.6) | 114 (5.6) | 165 (8) | 165 (14) | 82 (7) | 83 (7) |
| Unknown | 50 (2.4) | 20 (0.9) | 30 (1.5) | 30 (2.5) | 16 (1.3) | 14 (1.2) |
| **Breed Group** | **N = 2050** | | | **N = 1191** | | |
| Sporting | 457 (22.3) | 206 (10.1) | 251 (12.2) | 251 (21.1) | 145 (12.2) | 106 (8.9) |
| Mixed breed | 411 (20.1) | 178 (8.7) | 233 (11.4) | 233 (19.6) | 128 (10.8) | 105 (8.8) |
| Working | 225 (11) | 100 (4.9) | 125 (6.1) | 125 (10.5) | 68 (5.7) | 57 (4.8) |
| Toy | 195 (9.5) | 88 (4.3) | 107 (5.2) | 107 (9) | 67 (5.6) | 40 (3.4) |
| Hound | 184 (9) | 85 (4.2) | 99 (4.8) | 99 (8.3) | 58 (4.9) | 41 (3.4) |
| Terrier | 182 (8.9) | 53 (2.6) | 129 (6.3) | 129 (10.8) | 69 (5.8) | 60 (5) |
| Herding | 170 (8.3) | 56 (2.7) | 114 (5.6) | 114 (9.6) | 50 (4.2) | 64 (5.4) |
| Non-Sporting | 147 (7.2) | 53 (2.6) | 94 (4.6) | 94 (7.9) | 52 (4.4) | 42 (3.5) |
| Unknown | 79 (3.9) | 40 (2) | 39 (1.9) | 39 (3.3) | 22 (1.9) | 17 (1.4) |
| **Isolation source** | **N = 2050** | | | **N = 1191** | | |
| Abdominal cavity and fluid | 65 (3.2) | 26 (1.3) | 39 (1.9) | 39 (3.3) | 26 (2.2) | 13 (1.1) |
| Ear and Ocular | 112 (5.5) | 42 (2.1) | 70 (3.4) | 70 (5.9) | 43 (3.6) | 27 (2.3) |
| Feces | 96 (4.7) | 32 (1.6) | 64 (3.1) | 64 (5.4) | 36 (3) | 28 (2.4) |
| Respiratory tract | 80 (3.9) | 17 (0.8) | 63 (3.1) | 63 (5.3) | 27 (2.3) | 36 (3) |
| Skin | 30 (1.4) | 9 (0.4) | 21 (1) | 21 (1.8) | 13 (1.1) | 8 (0.7) |
| Urine and bladder | 1257 (61.3) | 584 (28.5) | 673 (32.8) | 673 (56.5) | 374 (31.4) | 299 (25.1) |
| Uterus, vagina, and vulva | 43 (2.1) | 23 (1.1) | 20 (1) | 20 (1.7) | 14 (1.2) | 6 (0.5) |
| Wounds | 52 (2.5) | 13 (0.6) | 39 (1.9) | 39 (3.3) | 17 (1.4) | 22 (1.9) |
| All others | 315 (15.4) | 113 (5.5) | 202 (9.9) | 202 (17) | 109 (9.2) | 93 (7.8) |

concerns about aminoglycoside toxicity. Similar to the results in amikacin, we found a near perfect susceptibility to imipenem suggesting that imipenem is rarely used in the treatment of bacterial diseases of dogs in Indiana. Imipenem belongs to the carbapenem antimicrobial class and is used in the treatment of multidrug resistant Enterobacteriaceae e.g. *E. coli* [25]. Perhaps this finding could reflect adherence by small animal clinicians to the guidelines for carbapenem use provided by the International Society for Companion Animal Infectious Diseases (ISCAID). The ISCAID recommends that carbapenems should be used only if the pathogen is proven to be resistant to all other reasonable antimicrobial options and susceptibility to the carbapenem chosen is documented [25].

In the present study, 61% of the *E. coli* isolates were found in specimens submitted from the urinary tract. This finding is similar to the findings in previous studies in the U.S. where most

**Table 5. Results of univariable logistic regression models assessing the association of host factors with antimicrobial resistance among *Escherichia coli* isolated from dog specimens originating from Indiana.**

| Host factors | Category | OR (95%CI) | P Value |
|---|---|---|---|
| Sex | [†]Overall | — | 0.6338 |
| | Male vs Female | 0.93 (0.76–1.13) | 0.442 |
| | Male vs Unknown | 1.04 (0.75–1.43) | 0.832 |
| | Female vs Unknown | 1.12 (0.82–1.52) | 0.473 |
| Age | [†]Overall | — | 0.0149 |
| | 1-3years vs >3-6years | 1.46 (1.02–2.1) | 0.039 |
| | 1-3years vs >6-8years | 1.85 (1.29–2.65) | 0.0008 |
| | 1-3years vs >8-10years | 1.54 (1.08–2.18) | 0.017 |
| | 1-3years vs >10-12years | 1.1 (0.76–1.59) | 0.614 |
| | 1-3years vs >12years | 1.34 (0.93–1.95) | 0.121 |
| | 1-3years vs Unknown | 1.3 (0.69–2.44) | 0.423 |
| | 1-3years vs <1year | 1.53 (1.01–2.31) | 0.043 |
| | >3-6years vs >6-8years | 1.27 (0.93–1.73) | 0.136 |
| | >3-6years vs >8-10years | 1.05 (0.78–1.42) | 0.750 |
| | >3-6years vs >10-12years | 0.75 (0.55–1.04) | 0.081 |
| | >3-6years vs >12years | 0.92 (0.66–1.27) | 0.606 |
| | >3-6years vs Unknown | 0.89 (0.48–1.63) | 0.695 |
| | >3-6years vs <1year | 1.05 (0.72–1.52) | 0.809 |
| | >6-8years vs >8-10years | 0.83 (0.62–1.12) | 0.218 |
| | >6-8years vs >10-12years | 0.59 (0.43–0.82) | 0.001 |
| | >6-8years vs >12years | 0.73 (0.53–1) | 0.05 |
| | >6-8years vs Unknown | 0.7 (0.38–1.28) | 0.248 |
| | >6-8years vs <1year | 0.83 (0.57–1.19) | 0.310 |
| | >8-10years vs >10-12years | 0.72 (0.53–0.94) | 0.034 |
| | >8-10years vs >12years | 0.87 (0.64–1.2) | 0.4 |
| | >8-10years vs Unknown | 0.84 (0.46–1.54) | 0.579 |
| | >8-10years vs <1year | 1 (0.7–1.43) | 0.986 |
| | >10-12years vs >12years | 1.22 (0.88–1.7) | 0.239 |
| | >10-12years vs Unknown | 1.18 (0.64–2.17) | 0.598 |
| | >10-12years vs <1year | 1.39 (0.96–2.03) | 0.085 |
| | >12years vs Unknown | 0.97 (0.52–1.78) | 0.909 |
| | >12years vs <1year | 1.14 (0.78–1.67) | 0.499 |
| | Unknown vs <1year | 1.18 (0.62–2.24) | 0.608 |
| Breed group | [†]Overall | — | 0.0007 |
| | Hound vs Mixed | 0.89 (0.63–1.26) | 0.512 |
| | Hound vs non-Sporting | 0.66 (0.42–1.02) | 0.064 |
| | Hound vs Sporting | 0.96 (0.68–1.35) | 0.797 |
| | Hound vs Terrier | 0.48 (0.31–0.74) | 0.0008 |
| | Hound vs Toy | 0.96 (0.64–1.44) | 0.835 |
| | Hound vs Unknown | 1.2 (0.71–2.03) | 0.509 |
| | Hound vs Working | 0.93 (0.63–1.38) | 0.723 |
| | Hound vs Herding | 0.57 (0.37–0.88) | 0.011 |
| | Mixed vs non-Sporting | 0.74 (0.5–1.09) | 0.126 |
| | Mixed vs Sporting | 1.07 (0.82–1.41) | 0.601 |
| | Mixed vs Terrier | 0.54 (0.37–0.78) | 0.001 |
| | Mixed vs Toy | 1.08 (0.76–1.52) | 0.673 |

(*Continued*)

**Table 5.** (Continued)

| Host factors | Category | OR (95%CI) | P Value |
|---|---|---|---|
| | Mixed vs Unknown | 1.34 (0.83–2.18) | 0.231 |
| | Mixed vs Working | 1.05 (0.76–1.45) | 0.783 |
| | Mixed vs Herding | 0.64 (0.44–0.94) | 0.021 |
| | Non-Sporting vs Sporting | 1.46 (0.99–2.14) | 0.055 |
| | Non-Sporting vs Terrier | 0.73 (0.46–1.16) | 0.182 |
| | Non-Sporting vs Toy | 1.46 (0.94–2.26) | 0.092 |
| | Non-Sporting vs Unknown | 1.82 (1.04–3.17) | 0.035 |
| | Non-Sporting vs Working | 1.42 (0.93–2.18) | 0.109 |
| | Non-Sporting vs Herding | 0.87 (0.55–1.39) | 0.561 |
| | Sporting vs Terrier | 0.5 (0.35–0.72) | 0.0002 |
| | Sporting vs Toy | 1 (0.72–1.4) | 0.99 |
| | Sporting vs Unknown | 1.25 (0.78–2.02) | 0.361 |
| | Sporting vs Working | 0.98 (0.71–1.34) | 0.876 |
| | Sporting vs Herding | 0.6 (0.41–0.87) | 0.006 |
| | Terrier vs Toy | 2 (1.31–3.07) | 0.001 |
| | Terrier vs Unknown | 2.5 (1.45–4.3) | 0.001 |
| | Terrier vs Working | 1.95 (1.29–2.95) | 0.002 |
| | Terrier vs Herding | 1.2 (0.76–1.88) | 0.439 |
| | Toy vs Unknown | 1.25 (0.74–2.11) | 0.408 |
| | Toy vs Working | 0.97 (0.66–1.43) | 0.888 |
| | Toy vs Herding | 0.6 (0.39–0.92) | 0.018 |
| | Unknown vs Working | 0.78 (0.47–1.3) | 0.343 |
| | Unknown vs Herding | 0.48 (0.28–0.83) | 0.008 |
| | Working vs Herding | 0.61 (0.41–0.93) | 0.02 |
| Sample source/sample type | [†]Overall | — | < .0001 |
| | Ear & ocular vs Feces | 0.83 (0.47–1.48) | 0.532 |
| | Ear & ocular vs Respiratory tract | 0.45 (0.23–0.87) | 0.017 |
| | Ear & ocular vs Skin | 0.71 (0.3–1.7) | 0.448 |
| | Ear & ocular vs Urine & bladder | 1.45 (0.97–2.15) | 0.069 |
| | Ear & ocular vs Uterus, vagina, vulva | 1.92 (0.94–3.9) | 0.073 |
| | Ear & ocular vs Wounds | 0.56 (0.27–1.16) | 0.117 |
| | Ear & ocular vs All others | 0.93 (0.6–1.46) | 0.758 |
| | Ear & ocular vs Abdominal cavity/fluid | 1.1 (0.59–2.08) | 0.742 |
| | Feces vs Respiratory tract | 0.54 (0.27–1.07) | 0.077 |
| | Feces vs Skin | 0.86 (0.35–2.09) | 0.734 |
| | Feces vs Urine & bladder | 1.74 (1.12–2.69) | 0.014 |
| | Feces vs Uterus, vagina, vulva | 2.3 (1.1–4.79) | 0.026 |
| | Feces vs Wounds | 0.67 (0.31–1.42) | 0.294 |
| | Feces vs All others | 1.12 (0.69–1.81) | 0.649 |
| | Feces vs Abdominal cavity/fluid | 1.3 (0.69–2.56) | 0.389 |
| | Respiratory tract vs Skin | 1.59 (0.62–4.09) | 0.338 |
| | Respiratory tract vs Urine & bladder | 3.2 (1.86–5.56) | < .0001 |
| | Respiratory tract vs Uterus, vagina, vulva | 4.26 (1.91–9.52) | 0.0004 |
| | Respiratory tract vs wounds | 1.24 (0.54–2.82) | 0.616 |
| | Respiratory tract vs all others | 2.07 (1.16–3.71) | 0.014 |
| | Respiratory tract vs Abdominal cavity/fluid | 2.47 (1.19–5.13) | 0.015 |
| | Skin vs Urine & bladder | 2.03 (0.92–4.46) | 0.08 |

(*Continued*)

**Table 5.** (Continued)

| Host factors | Category | OR (95%CI) | P Value |
|---|---|---|---|
| | Skin vs Uterus, vagina, vulva | 2.68 (1–7.18) | 0.049 |
| | Skin vs Wounds | 0.78 (0.29–2.12) | 0.623 |
| | Skin vs All others | 1.31 (0.58–2.95) | 0.521 |
| | Skin vs Abdominal cavity/fluid | 1.56 (0.62–3.92) | 0.349 |
| | Urine & bladder vs Uterus, vagina, vulva | 1.33 (0.72–2.44) | 0.365 |
| | Urine & bladder vs Wounds | 0.38 (0.2–0.73) | 0.003 |
| | Urine & bladder vs All others | 0.65 (0.5–0.83) | 0.0008 |
| | Urine & bladder vs Abdominal cavity/fluid | 0.77 (0.46–1.28) | 0.31 |
| | Uterus, vagina, vulva vs Wounds | 0.29 (0.12–0.69) | 0.005 |
| | Uterus, vagina, vulva vs All others | 0.49 (0.26–0.92) | 0.028 |
| | Uterus, vagina, vulva vs Abdominal cavity/fluid | 0.58 (0.27–1.26) | 0.17 |
| | Wounds vs All others | 1.68 (0.86–3.28) | 0.129 |
| | Wounds vs Abdominal cavity/fluid | 2 (0.9–4.45) | 0.09 |
| | All others vs Abdominal cavity/fluid | 1.19 (0.69–2.06) | 0.53 |

[†]Overall = overall effect of host factor on AMR.

of the *E. coli* were isolated from the urinary tract [16, 20]. This suggests that urinary tract infections could have been the major reason for canine sample submission to this laboratory. However, 3.7% of the *E. coli* isolates were from the respiratory tract and these respiratory tract isolates were more likely to be antimicrobial resistant than those isolated from the urogenital tract (urine, bladder, uterus, vagina, and vulva), and the abdominal cavity. This is in contrast to a previous study in the north eastern United States which reported that multidrug resistance was more likely among urinary *E. coli* than in *E. coli* isolated from other canine body sites [20]. *E. coli* is known to be involved in respiratory tract infections in dogs and has been isolated from respiratory tract samples [26, 27]. Possibly, the higher AMR observed in the respiratory tract isolates in our study could be due to selection pressure resulting from AMU targeting respiratory tract infections in these dogs. There is a need for an in-depth study of AMR among *E. coli* causing respiratory disease.

In the present study, we found that *E. coli* isolated from dogs older than 10 years were more likely to be resistant to antimicrobials when compared to *E. coli* isolated from younger dogs after controlling for breed and specimen source. This finding could be due to selection pressure from prior/routine antimicrobial use in dogs in this category since dogs older than 10 years are more likely to have been treated with antimicrobials multiple times when compared to younger dogs. Previous studies found prior use of antimicrobials was a risk factor for AMR in dogs [28, 29] and AMR *E. coli* was common among vet-visiting dogs [30]. Specifically, prior exposure to some antimicrobials such as fluoroquinolones may select for antimicrobial resistant *E. coli* in dogs that could persist long after antimicrobial therapy [31, 32]. Recurrent *E. coli* infections are possible because *E. coli* possess multiple adaptations for survival and persistence in the host [33]. Dogs older than 10 years are generally considered geriatric and are likely to have weakened immune systems due to old age, and as a result, could be susceptible to frequent infections necessitating antimicrobial use. Also, selection pressure from prior AMU could be the reason why isolates from dogs aged 1 to 3 years were 1.63 times more likely to be antimicrobial resistant when compared to those from dogs between 6 and 8 years of age. From a public health standpoint, the role of dogs aged older than 10 years and those aged 1 to 3 years in the dissemination of AMR *E. coli* needs to be further investigated. The implications are that

**Table 6. Results of univariable logistic regression models assessing the association of host factors with multi-drug resistance among *Escherichia coli* isolated from dog specimens originating from Indiana.**

| Host factors | Category | OR (95%CI) | P Value |
|---|---|---|---|
| Sex | [†]Overall | — | 0.4330 |
| | Male vs Female | 0.89 (0.69–1.15) | 0.378 |
| | Male vs Unknown | 1.12 (0.73–1.74) | 0.604 |
| | Female vs Unknown | 1.26 (0.84–1.9) | 0.269 |
| Age | [†]Overall | — | 0.2377 |
| | 1-3years vs >3-6years | 1.29 (0.83–2.01) | 0.267 |
| | 1-3years vs >6-8years | 1.21 (0.77–1.9) | 0.405 |
| | 1-3years vs >8-10years | 1.6 (1.03–2.47) | 0.035 |
| | 1-3years vs >10-12years | 1.13 (0.73–1.75) | 0.576 |
| | 1-3years vs >12years | 0.93 (0.59–1.47) | 0.761 |
| | 1-3years vs Unknown | 1.08 (0.49–2.38) | 0.852 |
| | 1-3years vs <1year | 0.96 (0.58–1.6) | 0.886 |
| | >3-6years vs >6-8years | 0.94 (0.62–1.44) | 0.78 |
| | >3-6years vs >8-10years | 1.24 (0.83–1.86) | 0.297 |
| | >3-6years vs >10-12years | 0.88 (0.59–1.32) | 0.537 |
| | >3-6years vs >12years | 0.72 (0.47–1.11) | 0.136 |
| | >3-6years vs Unknown | 0.84 (0.39–1.82) | 0.655 |
| | >3-6years vs <1year | 0.75 (0.46–1.22) | 0.248 |
| | >6-8years vs >8-10years | 1.32 (0.87–1.99) | 0.19 |
| | >6-8years vs >10-12years | 0.94 (0.62–1.41) | 0.749 |
| | >6-8years vs >12years | 0.77 (0.5–1.18) | 0.233 |
| | >6-8years vs Unknown | 0.89 (0.41–1.94) | 0.77 |
| | >6-8years vs <1year | 0.8 (0.48–1.31) | 0.366 |
| | >8-10years vs >10-12years | 0.71 (0.48–1.05) | 0.089 |
| | >8-10years vs >12years | 0.58 (0.39–0.88) | 0.011 |
| | >8-10years vs Unknown | 0.68 (0.31–1.46) | 0.318 |
| | >8-10years vs <1year | 0.6 (0.37–0.98) | 0.04 |
| | >10-12years vs >12years | 0.82 (0.54–1.25) | 0.357 |
| | >10-12years vs Unknown | 0.95 (0.44–2.06) | 0.901 |
| | >10-12years vs <1year | 0.85 (0.52–1.38) | 0.511 |
| | >12years vs Unknown | 1.16 (0.53–2.52) | 0.714 |
| | >12years vs <1year | 1.03 (0.63–1.7) | 0.899 |
| | Unknown vs <1year | 0.89 (0.39–2.02) | 0.786 |
| Breed group | [†]Overall | — | 0.3 |
| | Hound vs Mixed | 0.86 (0.54–1.39) | 0.54 |
| | Hound vs non-Sporting | 0.88 (0.5–1.55) | 0.647 |
| | Hound vs Sporting | 0.97 (0.6–1.55) | 0.889 |
| | Hound vs Terrier | 0.81 (0.48–1.38) | 0.443 |
| | Hound vs Toy | 1.18 (0.68–2.07) | 0.554 |
| | Hound vs Unknown | 0.92 (0.43–1.93) | 0.816 |
| | Hound vs Working | 0.84 (0.5–1.44) | 0.531 |
| | Hound vs Herding | 0.55 (0.32–0.95) | 0.033 |
| | Mixed vs non-Sporting | 1.02 (0.63–1.64) | 0.95 |
| | Mixed vs Sporting | 1.12 (0.78–1.61) | 0.53 |
| | Mixed vs Terrier | 0.94 (0.61–1.45) | 0.791 |
| | Mixed vs Toy | 1.37 (0.86–2.2) | 0.184 |
| | Mixed vs Unknown | 1.06 (0.54–2.1) | 0.864 |

*(Continued)*

**Table 6.** (Continued)

| Host factors | Category | OR (95%CI) | P Value |
|---|---|---|---|
| | Mixed vs Working | 0.98 (0.63–1.51) | 0.923 |
| | Mixed vs Herding | 0.64 (0.41–1.01) | 0.053 |
| | Non-Sporting vs Sporting | 1.11 (0.69–1.78) | 0.682 |
| | Non-Sporting vs Terrier | 0.93 (0.55–1.58) | 0.786 |
| | Non-Sporting vs Toy | 1.35 (0.77–2.38) | 0.294 |
| | Non-Sporting vs Unknown | 1.05 (0.49–2.22) | 0.908 |
| | Non-Sporting vs Working | 0.96 (0.56–1.65) | 0.892 |
| | Non-Sporting vs Herding | 0.63 (0.36–1.09) | 0.101 |
| | Sporting vs Terrier | 0.84 (0.55–1.29) | 0.426 |
| | Sporting vs Toy | 1.22 (0.77–1.95) | 0.393 |
| | Sporting vs Unknown | 0.95 (0.48–1.87) | 0.873 |
| | Sporting vs Working | 0.87 (0.57–1.34) | 0.535 |
| | Sporting vs Herding | 0.57 (0.37–0.89) | 0.014 |
| | Terrier vs Toy | 1.46 (0.86–2.46) | 0.158 |
| | Terrier vs Unknown | 1.13 (0.55–2.32) | 0.748 |
| | Terrier vs Working | 1.04 (0.63–1.7) | 0.884 |
| | Terrier vs Herding | 0.68 (0.41–1.13) | 0.135 |
| | Toy vs Unknown | 0.77 (0.37–1.63) | 0.497 |
| | Toy vs Working | 0.71 (0.42–1.21) | 0.207 |
| | Toy vs Herding | 0.47 (0.27–0.8) | 0.006 |
| | Unknown vs Working | 0.92 (0.45–1.9) | 0.826 |
| | Unknown vs Herding | 0.6 (0.29–1.26) | 0.177 |
| | Working vs Herding | 0.66 (0.39–1.09) | 0.104 |
| Sample source/sample type | [†]Overall | — | 0.1856 |
| | Ear & ocular vs Feces | 0.81 (0.41–1.61) | 0.543 |
| | Ear & ocular vs Respiratory tract | 0.47 (0.24–0.94) | 0.033 |
| | Ear & ocular vs Skin | 1.02 (0.37–2.78) | 0.969 |
| | Ear & ocular vs Urine & bladder | 0.79 (0.47–1.3) | 0.348 |
| | Ear & ocular vs Uterus, vagina, vulva | 1.47 (0.5–4.27) | 0.485 |
| | Ear & ocular vs Wounds | 0.49 (0.22–1.08) | 0.075 |
| | Ear & ocular vs All others | 0.74 (0.42–1.28) | 0.279 |
| | Ear & ocular vs Abdominal cavity/fluid | 1.26 (0.55–2.86) | 0.587 |
| | Feces vs Respiratory tract | 0.58 (0.29–1.18) | 0.132 |
| | Feces vs Skin | 1.26 (0.46–3.47) | 0.649 |
| | Feces vs Urine & bladder | 0.97 (0.58–1.63) | 0.917 |
| | Feces vs Uterus, vagina, vulva | 1.81 (0.62–5.32) | 0.278 |
| | Feces vs Wounds | 0.6 (0.27–1.34) | 0.214 |
| | Feces vs All others | 0.91 (0.52–1.61) | 0.749 |
| | Feces vs Abdominal cavity/fluid | 1.56 (0.68–3.56) | 0.296 |
| | Respiratory tract vs Skin | 2.18 (0.79–5.96) | 0.134 |
| | Respiratory tract vs Urine & bladder | 1.67 (0.99–2.81) | 0.055 |
| | Respiratory tract vs Uterus, vagina, vulva | 3.1 (1.06–9.15) | 0.039 |
| | Respiratory tract vs wounds | 1.03 (0.46–2.31) | 0.942 |
| | Respiratory tract vs all others | 1.56 (0.88–2.77) | 0.125 |
| | Respiratory tract vs Abdominal cavity/fluid | 2.67 (1.16–6.13) | 0.021 |
| | Skin vs Urine & bladder | 0.77 (0.32–1.88) | 0.566 |
| | Skin vs Uterus, vagina, vulva | 1.44 (0.39–5.27) | 0.586 |
| | Skin vs Wounds | 0.48 (0.16–1.42) | 0.179 |
| | Skin vs All others | 0.72 (0.29–1.82) | 0.488 |

(*Continued*)

**Table 6.** (Continued)

| Host factors | Category | OR (95%CI) | *P* Value |
|---|---|---|---|
| | Skin vs Abdominal cavity/fluid | 1.23 (0.41–3.71) | 0.713 |
| | Urine & bladder vs Uterus, vagina, vulva | 1.87 (0.71–4.91) | 0.207 |
| | Urine & bladder vs Wounds | 0.62 (0.32–1.18) | 0.147 |
| | Urine & bladder vs All others | 0.94 (0.68–1.29) | 0.686 |
| | Urine& bladder vs Abdominal cavity/fluid | 1.6 (0.81–3.17) | 0.178 |
| | Uterus, vagina, vulva vs Wounds | 0.33 (0.11–1.04) | 0.059 |
| | Uterus, vagina, vulva vs All others | 0.5 (0.19–1.36) | 0.175 |
| | Uterus, vagina, vulva vs Abdominal cavity/fluid | 0.86 (0.27–2.75) | 0.796 |
| | Wounds vs All others | 1.52 (0.76–3.01) | 0.237 |
| | Wounds vs Abdominal cavity/fluid | 2.59 (1.03–6.49) | 0.043 |
| | All others vs Abdominal cavity/fluid | 1.71 (0.83–3.51) | 0.146 |

[†]Overall = overall effect of host factor on AMR.

**Table 7. Multivariable binary logistic regression model of the associations between host factors and antimicrobial resistance among *Escherichia coli* isolated from samples from Indiana.**

| Host factors | Category | OR (95% CI) | *P* Value |
|---|---|---|---|
| Age | [†]Overall | — | 0.009 |
| | 1-3years vs >3-6years | 1.31 (0.9–1.9) | 0.159 |
| | 1-3years vs >6-8years | 1.63 (1.13–2.36) | 0.009 |
| | 1-3years vs >8-10years | 1.35 (0.94–1.94) | 0.103 |
| | 1-3years vs >10-12years | 0.89 (0.61–1.3) | 0.543 |
| | 1-3years vs >12years | 1.08 (0.73–1.59) | 0.718 |
| | 1-3years vs Unknown | 1.14 (0.59–2.2) | 0.697 |
| | 1-3years vs <1year | 1.5 (0.98–2.29) | 0.064 |
| | >3-6years vs >6-8years | 1.25 (0.91–1.72) | 0.167 |
| | >3-6years vs >8-10years | 1.04 (0.76–1.41) | 0.83 |
| | >10-12years vs >3-6years | 1.47 (1.06–2.05) | 0.023 |
| | >3-6years vs >12years | 0.82 (0.58–1.16) | 0.264 |
| | >3-6years vs Unknown | 0.87 (0.46–1.64) | 0.671 |
| | >3-6years vs <1year | 1.15 (0.78–1.69) | 0.493 |
| | >6-8years vs >8-10years | 0.83 (0.61–1.12) | 0.221 |
| | >10-12years vs >6-8years | 1.84 (1.33–2.55) | 0.0003 |
| | >12years vs >6-8years | 1.52 (1.09–2.12) | 0.014 |
| | >6-8years vs Unknown | 0.7 (0.37–1.31) | 0.26 |
| | >6-8years vs <1year | 0.92 (0.62–1.35) | 0.654 |
| | >10-12years vs >8-10years | 1.52 (1.11–2.1) | 0.009 |
| | >8-10years vs >12years | 0.8 (0.58–1.1) | 0.165 |
| | >8-10years vs Unknown | 0.84 (0.45–1.57) | 0.591 |
| | >8-10years vs <1year | 1.11 (0.76–1.62) | 0.598 |
| | >10-12years vs >12years | 1.21 (0.86–1.7) | 0.271 |
| | >10-12years vs Unknown | 1.28 (0.68–2.42) | 0.442 |
| | >10-12years vs <1year | 1.69 (1.13–2.51) | 0.01 |
| | >12years vs Unknown | 1.06 (0.56–2.01) | 0.858 |
| | >12years vs <1year | 1.39 (0.93–2.09) | 0.11 |
| | Unknown vs <1year | 1.31 (0.68–2.55) | 0.422 |

(*Continued*)

**Table 7.** (Continued)

| Host factors | Category | OR (95% CI) | P Value |
|---|---|---|---|
| Breed group | [†]Overall | — | 0.0007 |
| | Hound vs Mixed | 0.92 (0.64–1.31) | 0.632 |
| | Hound vs non-Sporting | 0.67 (0.42–1.05) | 0.081 |
| | Hound vs Sporting | 0.94 (0.66–1.35) | 0.749 |
| | Hound vs Terrier | 0.47 (0.3–0.73) | 0.0008 |
| | Hound vs Toy | 0.97 (0.64–1.46) | 0.865 |
| | Hound vs Unknown | 1.3 (0.75–2.25) | 0.343 |
| | Hound vs Working | 0.93 (0.62–1.41) | 0.742 |
| | Herding vs Hound | 1.68 (1.08–2.63) | 0.022 |
| | Mixed vs non-Sporting | 0.73 (0.49–1.08) | 0.114 |
| | Mixed vs Sporting | 1.03 (0.78–1.36) | 0.832 |
| | Terrier vs Mixed | 1.95 (1.33–2.86) | 0.0006 |
| | Mixed vs Toy | 1.05 (0.74–1.49) | 0.772 |
| | Mixed vs Unknown | 1.42 (0.87–2.34) | 0.165 |
| | Mixed vs Working | 1.02 (0.73–1.43) | 0.913 |
| | Mixed vs Herding | 0.65 (0.44–0.95) | 0.027 |
| | Non-Sporting vs Sporting | 1.42 (0.96–2.1) | 0.079 |
| | Non-Sporting vs Terrier | 0.71 (0.44–1.13) | 0.15 |
| | Non-Sporting vs Toy | 1.45 (0.93–2.28) | 0.104 |
| | Non-Sporting vs Unknown | 1.96 (1.11–3.47) | 0.021 |
| | Non-Sporting vs Working | 1.4 (0.91–2.17) | 0.127 |
| | Non-Sporting vs Herding | 0.89 (0.56–1.43) | 0.642 |
| | Terrier vs Sporting | 2.01 (1.38–2.93) | 0.0003 |
| | Sporting vs Toy | 1.02 (0.72–1.44) | 0.9 |
| | Sporting vs Unknown | 1.38 (0.84–2.26) | 0.2 |
| | Sporting vs Working | 0.99 (0.71–1.38) | 0.949 |
| | Sporting vs Herding | 0.63 (0.43–0.92) | 0.016 |
| | Terrier vs Toy | 2.06 (1.33–3.2) | 0.001 |
| | Terrier vs Unknown | 2.78 (1.59–4.86) | 0.0003 |
| | Terrier vs Working | 1.99 (1.3–3.06) | 0.002 |
| | Terrier vs Herding | 1.27 (0.8–2.01) | 0.317 |
| | Toy vs Unknown | 1.35 (0.79–2.32) | 0.275 |
| | Toy vs Working | 0.97 (0.65–1.44) | 0.872 |
| | Herding vs Toy | 1.62 (1.05–2.51) | 0.03 |
| | Unknown vs Working | 0.72 (0.42–1.22) | 0.22 |
| | Unknown vs Herding | 0.46 (0.26–0.8) | 0.006 |
| | Herding vs Working | 1.57 (1.03–2.4) | 0.037 |
| Sample source/sample type | [†]Overall | — | < .0001 |
| | Ear & ocular vs Feces | 0.8 (0.45–1.43) | 0.446 |
| | Respiratory tract vs Ear & ocular | 2.2 (1.12–4.34) | 0.023 |
| | Ear & ocular vs Skin | 0.9 (0.37–2.18) | 0.814 |
| | Ear & ocular vs Urine & bladder | 1.59 (1.06–2.39) | 0.026 |
| | Ear & ocular vs Uterus, vagina, vulva | 1.91 (0.92–3.94) | 0.081 |
| | Ear & ocular vs Wounds | 0.64 (0.3–1.35) | 0.24 |
| | Ear & ocular vs All others | 0.97 (0.61–1.53) | 0.889 |
| | Ear & ocular vs Abdominal cavity/fluid | 1.24 (0.66–2.35) | 0.504 |
| | Feces vs Respiratory tract | 0.57 (0.28–1.15) | 0.115 |

(*Continued*)

**Table 7.** (Continued)

| Host factors | Category | OR (95% CI) | *P* Value |
|---|---|---|---|
| | Feces vs Skin | 1.13 (0.46–2.79) | 0.795 |
| | Feces vs Urine & bladder | 1.99 (1.27–3.13) | 0.003 |
| | Feces vs Uterus, vagina, vulva | 2.39 (1.13–5.04) | 0.022 |
| | Feces vs Wounds | 0.8 (0.37–1.73) | 0.573 |
| | Feces vs All others | 1.22 (0.74–1.99) | 0.44 |
| | Feces vs Abdominal cavity/fluid | 1.56 (0.8–3.04) | 0.191 |
| | Respiratory tract vs Skin | 1.98 (0.75–5.2) | 0.166 |
| | Respiratory tract vs Urine & bladder | 3.5 (1.99–6.13) | < .0001 |
| | Respiratory tract vs Uterus, vagina, vulva | 4.2 (1.84–9.56) | 0.0006 |
| | Respiratory tract vs wounds | 1.41 (0.61–3.26) | 0.425 |
| | Respiratory tract vs all others | 2.13 (1.18–3.86) | 0.013 |
| | Respiratory tract vs Abdominal cavity/fluid | 2.74 (1.3–5.77) | 0.008 |
| | Skin vs Urine & bladder | 1.77 (0.79–3.94) | 0.165 |
| | Skin vs Uterus, vagina, vulva | 2.12 (0.78–5.77) | 0.141 |
| | Skin vs Wounds | 0.71 (0.26–1.96) | 0.511 |
| | Skin vs All others | 1.08 (0.47–2.46) | 0.861 |
| | Skin vs Abdominal cavity/fluid | 1.38 (0.54–3.54) | 0.499 |
| | Urine & bladder vs Uterus, vagina, vulva | 1.2 (0.64–2.24) | 0.566 |
| | Wounds vs Urine & bladder | 2.49 (1.3–4.74) | 0.006 |
| | Urine & bladder vs All others | 0.61 (0.47–0.8) | 0.0003 |
| | Urine& bladder vs Abdominal cavity/fluid | 0.78 (0.47–1.32) | 0.356 |
| | Wounds vs Uterus, vagina, vulva | 2.98 (1.24–7.2) | 0.015 |
| | Uterus, vagina, vulva vs All others | 0.51 (0.26–0.98) | 0.042 |
| | Uterus, vagina, vulva vs Abdominal cavity/fluid | 0.65 (0.3–1.44) | 0.29 |
| | Wounds vs All others | 1.52 (0.77–2.98) | 0.23 |
| | Wounds vs Abdominal cavity/fluid | 1.95 (0.87–4.37) | 0.107 |
| | All others vs Abdominal cavity/fluid | 1.28 (0.74–2.24) | 0.378 |

[†]Overall = overall effect of host factor on AMR.

humans in close contact with dogs in these age groups would be at a higher risk of exposure to AMR *E. coli*. Veterinarians should be made aware of the potential role of dogs aged older than10 years and those aged 1 to 3 years in the spread of AMR *E. coli*. Generally, owners of older dogs need to be aware of the AMR *E. coli* risk in older dogs and should be encouraged to observe infection prevention measures such as hand washing with soap and clean water after handling their animals.

The association between AMR and breed reported in this study is surprising. We found that terriers and herding dogs were more likely to harbor AMR *E. coli* when compared to other breed categories. This is an interesting finding that needs to be further investigated as no previous study has elucidated this.

One limitation of this study was the lack of data related the clinical history of the dogs from which samples were collected. This prevented us from discerning the severity of the disease the dog presented with. Further, the lack of specific information regarding prior antimicrobial use in the dogs included in the study limits the inferences that can be made regarding AMU and its relationship with subsequent development of AMR.

## Conclusions

Our findings suggest that AMR in *E. coli* in dogs could be increasing in the state of Indiana. Dogs aged more than 10 years and those aged 1 to 3 years could play a role in the spread of AMR. *E. coli* in dogs in Indiana are likely to be highly susceptible to aminoglycosides (e.g., amikacin) and to carbapenems (e.g., imipenem). The findings of this study should inform efforts aimed at addressing the AMR challenge and may prove useful in guiding small animal clinicians in the state of Indiana in choosing appropriate antimicrobials for empiric therapy.

## Supporting information

**S1 File.**
(DOCX)

## Acknowledgments

We thank the Indiana Animal Disease Diagnostic Laboratory for providing the data used in these analyses.

## Author Contributions

**Conceptualization:** John E. Ekakoro, Lynn F. Guptill, Audrey Ruple.

**Data curation:** John E. Ekakoro, G. Kenitra Hendrix, Audrey Ruple.

**Formal analysis:** John E. Ekakoro, Audrey Ruple.

**Funding acquisition:** Audrey Ruple.

**Investigation:** John E. Ekakoro, Audrey Ruple.

**Methodology:** John E. Ekakoro, Audrey Ruple.

**Project administration:** Audrey Ruple.

**Resources:** G. Kenitra Hendrix, Audrey Ruple.

**Software:** Audrey Ruple.

**Supervision:** Audrey Ruple.

**Validation:** Audrey Ruple.

**Visualization:** Audrey Ruple.

**Writing – original draft:** John E. Ekakoro, Audrey Ruple.

**Writing – review & editing:** G. Kenitra Hendrix, Lynn F. Guptill, Audrey Ruple.

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
