## [Decision Letter · Decision Letter 0]

3 Mar 2022

PONE-D-22-02925Antimicrobial susceptibility and risk factors for resistance among Escherichia coli isolated from canine specimens submitted to a diagnostic laboratory in Indiana, 2010-2019PLOS ONE

Dear Dr. Ruple,

Thank you for submitting your manuscript to PLOS ONE. After careful consideration, we feel that it has merit but does not fully meet PLOS ONE’s publication criteria as it currently stands. Therefore, we invite you to submit a revised version of the manuscript that addresses the points raised during the review process.

Please address the questions raised by the reviewers. 

We look forward to receiving your revised manuscript.

Kind regards,

Iddya Karunasagar

Academic Editor

PLOS ONE

Journal Requirements:

3. Please include a complete copy of PLOS’ questionnaire on inclusivity in global research in your revised manuscript. Our policy for research in this area aims to improve transparency in the reporting of research performed outside of researchers’ own country or community. The policy applies to researchers who have travelled to a different country to conduct research, research with Indigenous populations or their lands, and research on cultural artefacts. The questionnaire can also be requested at the journal’s discretion for any other submissions, even if these conditions are not met.  Please find more information on the policy and a link to download a blank copy of the questionnaire here: https://journals.plos.org/plosone/s/best-practices-in-research-reporting. Please upload a completed version of your questionnaire as Supporting Information when you resubmit your manuscript.

"We thank the Integrative Data Science Initiative at Purdue University for funding this work and the Indiana Animal Disease Diagnostic Laboratory for providing the data used in these analyses."

"The Integrative Data Science Initiative at Purdue University (https://www.purdue.edu/data-science/) provided funding to AR for this work and JE was provided salary using these funds. The funders had no role in study design, data collection and analysis, decision to publish, or preparation of the manuscript."

Additional Editor Comments:

The reviewer has asked for some clarifications. Please revise addresseg all reviewer comments.

Reviewers' comments:

Reviewer's Responses to Questions

**Comments to the Author**

1. Is the manuscript technically sound, and do the data support the conclusions?

Reviewer #1: Yes

2. Has the statistical analysis been performed appropriately and rigorously? 

Reviewer #1: Yes

3. Have the authors made all data underlying the findings in their manuscript fully available?

Reviewer #1: Yes

4. Is the manuscript presented in an intelligible fashion and written in standard English?

Reviewer #1: Yes

5. Review Comments to the Author

Reviewer #1: Manuscript:# Antimicrobial susceptibility and risk factors for resistance among Escherichia coliisolated from canine specimens submitted to a diagnostic laboratory in Indiana, 2010-2019.

The authors calculated the proportion of antimicrobial susceptible E. coli isolated from canine specimens and identified their temporal patterns of susceptibility from 2010 through 2019. The overall percentage of AMR isolates was 61.7% and 29.3 % of isolates were multidrug resistant. The authors concluded that the proportion of susceptible isolates to several beta-lactam antimicrobials could be decreasing. Overall this is a good study and presented well

Comments:

How the isolates were designated as MDR?

Mention the name of the manufacturer of the antibiotic discs used in the study.

Provide reference for CLSI guidelines (year)

Which strain of bacteria was used as a control strain in antimicrobial testing?

Describe the media used for carrying out AST.

6. PLOS authors have the option to publish the peer review history of their article (what does this mean?). If published, this will include your full peer review and any attached files.

Reviewer #1: No

---

## [Author Response · Author response to Decision Letter 0]

29 Mar 2022

The authors calculated the proportion of antimicrobial susceptible E. coli isolated from canine specimens and identified their temporal patterns of susceptibility from 2010 through 2019. The overall percentage of AMR isolates was 61.7% and 29.3 % of isolates were multidrug resistant. The authors concluded that the proportion of susceptible isolates to several beta-lactam antimicrobials could be decreasing. Overall, this is a good study and presented well

Response: We thank you so much for reviewing our paper. Your comments have helped us improve the paper. Thank you again.

Comments:

How were the isolates designated as MDR?

Response: The isolates were designated as MDR based on the method described by Michael T sweeney and others (Sweeney, Michael T., et al. "Applying definitions for multidrug resistance, extensive drug resistance and pandrug resistance to clinically significant livestock and companion animal bacterial pathogens." Journal of Antimicrobial Chemotherapy 73.6 (2018): 1460-1463.). We have provided a citation [reference 13] of this paper by Sweeney et al, in the revised manuscript (please see line 136 of the revised manuscript with track changes). Thank you.

Mention the name of the manufacturer of the antibiotic discs used in the study.

Response: The antimicrobial susceptibility testing on the bacterial isolates was done using the broth microdilution method. We did not use the disc diffusion method, and hence are unable to mention the name of the manufacturer of the antibiotic discs. In our initial manuscript submission, we had not specifically mentioned that we used the broth microdilution method for antimicrobial susceptibility testing, although we mentioned using a dilution-based method. We now specifically mention this in our revised manuscript, and we also mention that we used the Sensititre™ Companion Animal Gram Negative COMPGN1F Vet AST Plates purchased from ThermoFisher scientific-USA. Please see lines 79-84 in the revised manuscript with track changes. Thank you.

Provide reference for CLSI guidelines (year)

Response: We have revised lines 86-87 (please see lines 86-87 in the revised manuscript with track changes) to mention that the Clinical and Laboratory Standards Institute (CLSI) guidelines used were those that were current at the time the isolate was tested. We have provided reference 6 for the CLSI guidelines (2018) which also mentions the previous editions of the VET08 CLSI guidelines.

Which strain of bacteria was used as a control strain in antimicrobial testing?

Response: Escherichia coli (ATCC® 25922™) was used as the quality control strain. We mention this in our revised manuscript (please see lines 79-84 in the revised manuscript with track changes). Thank you. 

Describe the media used for carrying out AST.

Response: The media used was the Mueller-Hinton broth. We mention this in our revised manuscript (please see lines 79-84 in the revised manuscript with track changes). Thank you.

---

## [Decision Letter · Decision Letter 1]

8 Aug 2022

Antimicrobial susceptibility and risk factors for resistance among Escherichia coli isolated from canine specimens submitted to a diagnostic laboratory in Indiana, 2010-2019

PONE-D-22-02925R1

Dear Dr. Ruple,

We’re pleased to inform you that your manuscript has been judged scientifically suitable for publication and will be formally accepted for publication once it meets all outstanding technical requirements.

Kind regards,

Iddya Karunasagar

Academic Editor

PLOS ONE

Additional Editor Comments (optional):

All reviewer comments have been addressed.

Reviewers' comments:

Reviewer's Responses to Questions

**Comments to the Author**

1. If the authors have adequately addressed your comments raised in a previous round of review and you feel that this manuscript is now acceptable for publication, you may indicate that here to bypass the “Comments to the Author” section, enter your conflict of interest statement in the “Confidential to Editor” section, and submit your "Accept" recommendation.

Reviewer #1: All comments have been addressed

2. Is the manuscript technically sound, and do the data support the conclusions?

Reviewer #1: Yes

3. Has the statistical analysis been performed appropriately and rigorously? 

Reviewer #1: N/A

4. Have the authors made all data underlying the findings in their manuscript fully available?

Reviewer #1: Yes

5. Is the manuscript presented in an intelligible fashion and written in standard English?

Reviewer #1: Yes

6. Review Comments to the Author

Reviewer #1: The authors addressed the points raised in the previous review. The manuscript is well presented. I do not have further comments

7. PLOS authors have the option to publish the peer review history of their article (what does this mean?). If published, this will include your full peer review and any attached files.

Reviewer #1: No

---

## [Editor Report · Acceptance letter]

15 Aug 2022

PONE-D-22-02925R1 

Antimicrobial susceptibility and risk factors for resistance among *Escherichia coli* isolated from canine specimens submitted to a diagnostic laboratory in Indiana, 2010-2019 

Dear Dr. Ruple:

I'm pleased to inform you that your manuscript has been deemed suitable for publication in PLOS ONE. Congratulations! Your manuscript is now with our production department. 

Kind regards, 

on behalf of

Dr. Iddya Karunasagar 

Academic Editor

PLOS ONE